# Design and Characterization of Lipid-Surfactant-Based Systems for Enhancing Topical Anti-Inflammatory Activity of Ursolic Acid

**DOI:** 10.3390/pharmaceutics15020366

**Published:** 2023-01-21

**Authors:** Bruno Fonseca-Santos, Giovanna Angeli Araujo, Paula Scanavez Ferreira, Francesca Damiani Victorelli, Andressa Maria Pironi, Victor Hugo Sousa Araújo, Suzana Gonçalves Carvalho, Marlus Chorilli

**Affiliations:** 1Rector Miguel Calmon Avenue, Department of Biotechnology, Health Sciences Institute, Federal University of Bahia (UFBA), Campus of Canela, Salvador 40110-902, BA, Brazil; 2School of Pharmaceutical Sciences, São Paulo State University—UNESP, Araraquara 14800-903, SP, Brazil

**Keywords:** ursolic acid, skin, topical product, inflammation, bioadhesion, surfactant

## Abstract

Skin inflammation is a symptom of many skin diseases, such as eczema, psoriasis, and dermatitis, which cause rashes, redness, heat, or blistering. The use of natural products with anti-inflammatory properties has gained importance in treating these symptoms. Ursolic acid (UA), a promising natural compound that is used to treat skin diseases, exhibits low aqueous solubility, resulting in poor absorption and low bioavailability. Designing topical formulations focuses on providing adequate delivery via application to the skin surface. The aim of this study was to formulate and characterize lipid-surfactant-based systems for the delivery of UA. Microemulsions and liquid crystalline systems (LCs) were characterized by polarized light microscopy (PLM), rheology techniques, and textural and bioadhesive assays. PLM supported the self-assembly of these systems and elucidated their formation. Rheologic examination revealed pseudoplastic and thixotropic behavior appropriate, and assays confirmed the ability of these formulations to adhere to the skin. In vivo studies were performed, and inflammation induced by croton oil was assessed for response to microemulsions and LCs. UA anti-inflammatory activities of ~60% and 50% were demonstrated by two microemulsions and 40% and 35% by two LCs, respectively. These data support the continued development of colloidal systems to deliver UA to ameliorate skin inflammation.

## 1. Introduction

Ursolic acid (UA) is a type of natural pentacyclic triterpenoid carboxylic acid found mainly in the peels of fruits [1,2] and in spices such as rosemary [3,4]. UA has numerous pharmacological effects, including antioxidant, anti-inflammatory, antibacterial, and antifungal properties [5,6,7,8]. Due to its poor solubility in water, the incorporation of UA into alternate drug delivery systems has been explored and found to be highly efficient to improve its solubility [9], such as with polymeric nanoparticles [10], liposomes [11], polymeric micelles [12], and nanostructured lipid carriers [13].

Drug delivery systems have been widely studied for the improvement of drug administration to the skin [14]. Surfactant-based systems including microemulsions and liquid crystals comprise amphiphilic surfactants that dissolve in water and oil and self-assemble into various ordered structures [15]. Microemulsions are clear, thermodynamically stable isotropic liquids in which the dispersed phase takes the form of very small droplets [16,17,18]. A liquid crystal is a state of matter that exhibits both the ordered properties of solids and the flow characteristics of liquids [19,20,21,22,23,24,25]. Microemulsions, as well as lamellar, hexagonal, or cubic shaped mesophases, have attracted attention in the pharmaceutical field for their physicochemical properties, such as viscosity [26], bioadhesion, ability to control drug release [24,27], and loading of hydrophilic or lipophilic drugs [28,29,30].

Skin inflammation and rashes can cause redness, pain, itching, and dryness [31,32]. Common inflammatory skin conditions include dermatitis, psoriasis, poison ivy and poison oak, and drug rashes [32,33,34,35,36]. Unfortunately, the resilient barrier presented by epidermal layers of the skin, primarily the stratum corneum, severely limits the efficacy of dermal and transdermal drug delivery and inhibits the effective diffusion of drugs [37].

The aim of this study was to develop and characterize a surfactant-based system to improve topical anti-inflammatory drug delivery, and in addition to evaluate the biological performance of the drug delivery system.

## 2. Materials and Methods

### 2.1. Materials

Oleic acid (OA) was obtained from Synth (Diadema, Brazil), Procetyl^®^ AWS (PPG-5-CETETH-20) was purchased from Croda (São Paulo, Brazil), and ursolic acid was obtained from Idealfarma (Anápolis, Brazil). Croton oil was purchased from Sigma-Aldrich (St. Louis, MO, USA). Purified water (W) was obtained by Milli-Q^®^ Millipore™ Direct 8/16 system (Burlington, MA, USA). All other reagents were of analytical grade and were used as received without further purification.

### 2.2. Ternary Phase Diagram

The ternary phase diagrams were constructed by mixing the oil (O), OA, PPG-5-CETETH-20 surfactant (S), and purified W. Weighted proportions of each component were varied in 10% increments, bringing the ternary mixture to a total of 100%. All components were mixed and stirred by vigorous manual shaking. After 48 h, with systems at phase equilibria, they were classified by macroscopic appearance.

### 2.3. Selected Formulations and Incorporation of Ursolic Acid 

Four formulations were selected based on ternary phase diagrams and designated A, B, C, and D. Component proportions (OA%/S%/W%) were 40/50/10, 30/50/20, 20/50/30, and 10/50/40 for formulation A, B, C, and D, respectively.

A solubility test of UA in OA was performed prior to incorporation into the formulations, prompting a target concentration of 5 mg/g. First, the UA was pulverized in a glass mortar with aid of a glass pestle in order to increase the contact surface of the particle and improve its solubility in the system. Next, OA was added, homogenizing until drug solubilization, then PPG-5-CETETH-20 was added and mixed. 

The formulations were incubated for ten minutes in an Elmasonic E 120 H ultrasound bath (Singen, Germany) at 37 kHz and room temperature for total solubilization of UA. Finally, W was added to the system and mixed with vigorous manual agitation. Formulations omitting UA were prepared in the same manner, excluding the drug comminution and ultrasound steps.

### 2.4. Polarized Light Microscopy

PLM was performed using a Zeiss Axioskop 40 (Thornwood, NY, USA). A drop of each formulation was placed on a microscopy slide and covered with a cover slip. Images were acquired using a digital camera (Samsung DV-70, Suwon-si, Republic of Korea) at 40× magnification.

### 2.5. Flow Rheology 

Flow curves were generated using a Discovery HR-1 Hybrid Rheometer (TA Instruments, New Castle, DE, USA) attached to a cone-plate geometry (∅ = 40 mm and θ = 2°) with a 52 μm gap. The sample was applied to base plate with minimal shear and fully filled the gap between plate and geometry. Experiments used a shear rate range of 0.1 to 100 (1/s) for the ascending curve and from 100 to 0.1 (1/s) for the descending curve, with each curve running for 120 s.

Quantitative analysis of flow behavior was calculated using the Herschel–Bulkley model with Equation (1) allowing determination of the consistency index (k) and the flow index (n):(1)$$\sigma =\sigma_{o}{\;+\;k\;\times \;\gamma }^{n}$$
where *σ* is shear stress (Pa), *σ_o_* is yield shear stress (Pa), and γ is shear rate (1/s).

### 2.6. Oscillatory Rheology 

Analysis of rheological behavior was performed using a Discovery HR-1 Hybrid Rheometer (TA Instruments, New Castle, DE, USA), using a cone-plate geometry (∅ = 40 mm and θ = 2°) with 52 μm gap. The sample was carefully applied to the base plate and the space between plate and cone was fully filled with samples. The linear viscosity range was determined at 1 Pa over a frequency range of 0.1 to 10 Hz. All analyses were performed at 32 °C in triplicate.

Quantitative data correlating the dependence of the storage modulus (G′) to the frequency were calculated by Equation (2):(2)$${G^{\prime }=S\;\times \;\omega }^{n}$$
where G′ is the storage modulus (Pa), ω is the frequency (Hz), S is the gel strength, and n is the viscoelastic exponent.

### 2.7. Texture Profile Analysis

Eight grams of each selected sample were placed in Falcon^®^ tubes (50 mL) and centrifuged for 3 min at 2663× *g* to ensure a smooth surface and the elimination of air bubbles. The tubes were left to rest for 24 h and subsequently analyzed using a texture analyzer; TA.XTplus (Stable Micro Systems, Godalming, UK).

The sample was placed beneath a cylindrical analytical probe (∅ = 10 mm) and compression was initiated at a velocity of 0.5 mm/s. When the probe reached 10 mm of depth the probe was raised out of the sample at 0.1 mm/s. After 5 s, the probe ran a second cycle at the same parameters. All analyses were performed at 25 °C and each sample was analyzed in triplicate.

Graphs of force by time were acquired and peaks of hardness and areas under curve (AUC) of compressibility and adhesiveness were achieved using Exponent^®^ (Stable Micro Systems Data Analysis Godalming, UK). Cohesion was calculated as the difference of compressibility area between the second and first cycle.

### 2.8. Bioadhesion Studies

Pig ears were obtained from a local slaughterhouse and pre-treated according to Dick and Scott [38] for subsequent analyses. Ears in good condition were sanitized with water at room temperature and the skin was separated from the cartilage with the aid of a scalpel. Ears that presented some type of injury such as wounds or minor lesions were discarded.

After sorting and pre-treatment, a layer of 400 μm was separated from the adipose layer using the Nouvag^®^ TCM 300 dermatometer (Goldach, Switzerland), retaining only the stratum corneum and epidermis. Any existing hairs were cut, and the specimens were kept in 0.9% sodium chloride solution for 30 min.

An 8 g sample of each selected formulation was placed in a Falcon^®^ tube (50 mL) and centrifuged for 3 min at 2663× *g*. The tubes were left to rest for 24 h and analyzed using texture analyzer TA.XT*plus* (Stable Micro Systems). On the day of the experiment, the tubes containing the samples were kept in a thermostatic water bath at 32 °C.

Using a cylindrical analytical probe (∅ = 10 mm), an ear skin specimen was attached to the bottom of the probe and secured with a rubber band. The probe was set to descend at a constant velocity of 1 mm/s and the sample was maintained in contact with the skin for a period of 60 s. During the contact stage, the probe was maintained at a constant depth and at end the probe was set to rise at 0.5 mm/s. Skin contact was interrupted and the work and force required to separate was calculated using Exponent^®^ Stable Micro Systems Data Analysis.

### 2.9. Croton-Oil-Induced Ear Edema

#### 2.9.1. Animals

Male Swiss mice (*Mus musculus*), 25–30 g, were housed at an average temperature of 22 °C (±2 °C) and kept in a 12 h light/dark cycle with feed and water ad libitum. The experimental protocol was approved by the Ethics Committee of the Use of Animals at the School of Pharmaceutical Sciences (CEUA-FCF) under protocol number 46/2018 in accordance with the guidelines, procedures, and bioethics of animal procedures from the Brazilian Council for Control of Animal Experimentation (CONCEA) standards.

#### 2.9.2. Treatments 

Mice were randomized to 9 groups (*n* = 6) and all treatments were administered to the right ear. Group 1 were naïve mice that received no induction nor any treatment. Group 2 received no treatment (negative control); group 3 received formulation A without UA. Groups 4–7 were administered formulations A to D loaded with AU (5 mg/g), respectively; group 8 received UA solubilized in OA (5 mg/g); and group 9 was administered dexamethasone cream (DEX) (0.1 mg/g).

#### 2.9.3. Edema Induced by Croton Oil

The ear edema was induced on the external right ear by topical application of 20 µL of croton oil in acetone (50 µg/mL, 1 µg/ear). At 30 min prior to the induction, 20 mg of drug formulation was applied topically to the inner right ear of the animals according to the groups outlined above. Ear edema was measured both before and 6 h after induction of inflammation. Animals were euthanized, the ears hole-punched, and the isolated specimens were weighed. Ear edema was quantified as the difference in weight (in mg) of the section removed from the right ear (treated with phlogistic agent, formulations, and positive control) and the weight (in mg) of the section removed from the left ear (naïve group).

The average percent inhibition of edema (%) was calculated using the following Equation (3):(3)$$\mathrm{Inhibition}\;\mathrm{of}\;\mathrm{edema}\;\left(\%\right)=1-\frac{\left(NC-NV\right)}{\left(T-NV\right)}\times 100\;$$
where *NC* is negative control group; *NV* is naïve group; and *T* is tested samples.

### 2.10. Statistical Analysis

When applicable, significant differences between the means of the values obtained were submitted to analysis of variance (ANOVA) followed by multiple comparisons applying the Tukey method (*p* < 0.05).

## 3. Results and Discussion

A ternary phase diagram was obtained for oil, water, and surfactant mixtures and is shown in Figure 1a. Higher proportions of surfactant lead to the formation of isotropic liquid systems, such as micelles and microemulsions, observable at the upper vertex of the diagram. However, when the water content increased, it undermined the clear gelling systems and led to characteristic hexagonal or cubic mesophases. The ternary phase diagram shows these vertices in two regions: the left side, characterized by opaque emulsion-like systems, and a phase separation region on the right side.

Surfactants are known to self-assemble to several types of colloidal systems, microemulsions, macroemulsions, and lyotropic liquid crystals [17,18,39,40], and these colloidal systems have been examined for topical drug delivery to skin [41,42,43,44,45,46,47,48,49]. The formation of microemulsions and lyotropic liquid crystals is dependent on several factors, including temperature, salt concentrations, hydrophilic–lipophilic balance (HLB) of surfactant, water solvation, and supramolecular interactions between the components [50]. Recently, the critical packing parameter (CPP) of surfactants has been used to explain these self-assemblies and the behavior of mixtures undergoing the phase change to lyotropic liquid crystals [51,52].

The CPP is a geometric parameter of a surfactant defined as volume of the tail chain (*v_s_*) divided by the product of the area of the head group at the head–tail interface (*a*_0_) and critical length of the tail chain (*l_c_*) [51], as shown in Equation (4). Thus, the aggregation shape of amphiphilic molecules can be determined according to the value of CPP. For example, from 0 < CPP ≤ 1/3 only spherical micelles exist in solution [51]. However, when 1/3 < CPP ≤ 1/2, aggregations with a cylindrical micelle or worm-like micelle are most likely. Finally, for 1/2 < CPP ≤ 1, there is a balance between sizes of the head group and tail, which causes the surfactant molecule to form planer aggregates with a bi-layer structure [51].
(4)CPP=vsa0×lc 

Water content is an influencing factor of phase transition, and with the decreasing water content, different structures are derived, such as the normal micelle (oil in water), the normal cubic phase, the normal hexagonal phase, the lamellar phase, the reversed cubic phase (water in oil), and the reversed hexagonal phase [50]. In our case, dilution of water in the ternary phase system shows a transition from the isotropic region to a viscous isotropic region, which characterizes lyotropic liquid crystalline systems. In this instance, water molecules solvate the head group of surfactants, changing the negative curvature and leading to the generation of lyotropic liquid crystals.

The selected formulations chosen all represented 50% (*w*/*w*) of the surfactant and oil/water ratios with variations shown to correspond with the inner area of the ternary phase diagram (Figure 1a). Four formulations were carefully chosen, with two selected due to their liquid macroscopic appearance and two selected due to gel-like viscosity. Formulations were analyzed by PLM: the liquid formulations (named A and B) displayed isotropic-like properties and a dark field was seen under PLM. Alternatively, formulations C and D showed anisotropic properties under PLM and were characterized as liquid crystalline systems (Figure 1b). PLM is a characterization technique and must be used to accurately categorize LCs [53]; isotropic mixtures show a dark field distinctive for microemulsions [43,54] and cubic mesophases [55]. These two systems are then distinguishable by viscosity, with microemulsions presenting viscosity behavior similar to water [17], while cubic mesophases are categorized as a firm clear gel. Anisotropy is characteristically used for classification of lyotropic LCs [53], while lamellar mesophases display Maltese’s crosses [44,47] and hexagonal mesophases present as stretch marks under PLM [46]. Formulations A and B were characterized as microemulsions, while C and D proved to be hexagonal mesophases. The addition of AU did not change the polarizing appearance of samples, as shown in Figure 1b.

Formulations A to D were analyzed by flow and oscillatory rheology (Figure 2). Flow rheology of all formulations showed a shear-thinning behavior, revealing fluids whose viscosity decreases under shear strain, as well as a hysteresis region revealing thixotropic fluid behavior. For loaded and unloaded formulations A and B, a low viscosity was exhibited, while C and D formulations demonstrated high viscosity values. For *n* < 1, the fluid is shear-thinning, whereas for *n* > 1 the fluid is shear-thickening. If *n* = 1 and the yield stress rate is 0, this model reduces to the Newtonian fluid. The Herschel–Bulkley model characterizes A and B formulations with n values close to 1 and no yield stress rate values, classifying them as Newtonian fluids, with the exception of sample AAU which presented a yield stress rate and was classified as ideal Bingham plastic (Table 1). All other samples presented *n* < 1 (values between 0.8 and 0.3) and were classified as pseudoplastic or shear-thinning fluids. Only samples C and D, characterized as hexagonal liquid crystals, were able to exhibit that a minimum stress is required to break down, to some extent, the internal ‘structure’ of the fluid before any movement (yield stress rates exhibited in Table 1). Samples characterized as microemulsions (A and B) do not show this behavior, explained by their ‘weak’ structure.

Upon application of shear force, the sample undergoes disorganization of molecules, resulting in low viscosity. For the application of topical products to skin, this is analogous to thinning of the product at the high shear rates typical of application [56]. Thixotropy provides an indication of the degree to which the product flows into the skin surface. Quick recuperation of viscosity in a topical product leads the formulation to remain on the skin, giving a greasy feel. Shear thinning and moderate thixotropic characteristics in a topical product represent very favorable rheological behaviors for providing uniform coverage of the skin surface [56]. To this end, skin care products and cosmetics have been formulated to exbibit pseudoplastic behavior, and among these products, we can refer to examples of emulsions and lotions [57,58,59], microemulsions [60,61], hydrogels [61,62,63], and lyotropic liquid crystals [45,64,65]. Previous studies performed by our group demonstrated that lyotropic liquid crystals designed with other surfactants and natural oils are ideal for formulating Non-Newtonian fluids, showing pseudoplastic behavior and thixotropic characteristics [46,47,66].

The consistency index (k), shown in Table 1, reveals higher values when a drug is added to the systems. In addition, when internal water is increased in the system, k values are also high, suggesting internal supramolecular self-assembled formulations. K values have been studied to characterize polymers and polymer solutions, with increasing polymer concentration in solution or cross-linking of polymeric chains generally leading to higher k values [67,68]. This parameter is also directly related to the viscosity of the material [69].

In oscillatory rheology, the storage modulus, G′, represents the elastic portion of viscoelastic behavior, which describes the solid-state behavior of the sample. Meanwhile, the loss modulus, G″, characterizes the viscous portion of viscoelastic behavior, or the liquid-state behavior of the sample [70]. Viscoelastic solids have a higher storage modulus than loss modulus (G′ > G″) due to links inside the material, for example chemical bonds or physical–chemical interactions [71]. On the other hand, viscoelastic liquids with a higher loss modulus than storage modulus (G″ > G′) [72,73] generally have no such strong bonds between the individual molecules. The addition of drugs changed this behavior in formulation B (BAU), where this sample showed a more structured system. Alternately, higher values of S have been related to density structures, molecular interconnections, or the packing of molecules [74]. The use of the Power Law has been studied to predict the viscoelastic behavior of hydrogels [75,76,77]. The use of this technique has also been applied to the study of colloidal systems, such as microemulsions and lyotropic liquid crystals [78,79]. For a quantitative analysis, a G′ modulus value is fitted by the Power Law model to give the S constant (relating to gel strength) and n as the viscoelastic exponent. These values are shown in Table 1. Note the value of n decreases with increasing cross-linking density due to microstructure. 

For samples characterized as microemulsions, these moduli correspond to G″ > G′, and they show a ‘weak’ microstructure of material. When water was added to these systems, it resulted in a self-assembly to lyotropic liquid crystals and a switch to G′ > G″. Graphs (Figure 2c,d) show frequency-dependent sweeps, that is, with increasing frequency there is a deformation or breakdown of the material, and it is possible to weaken bonds and the interactions of microemulsions. However, when the material is structured, no changes are exhibited and even with increasing frequency the stored energy is constant. These behaviors have been described for micelles, microemulsions, or lamellar mesophases with a G″ > G′ [42,78] and hexagonal and cubic mesophases denoted by a G′ > G″ [64,65].

TPA is a technique for evaluating the textural properties of semisolid products, and this technique allows measurement of hardness, adhesiveness, compressibility, and cohesiveness (Figure 3) [80]. Hardness is related to the maximum force required to rupture the matrix, adhesiveness is the sample’s force under the probe surface, compressibility is the force to deform the semisolid product under first compression, and cohesiveness is related to the ratio of areas under the curve of the second and first cycles of compression [81].

Due to the unsuitability of liquid samples in this analysis, liquid samples were not analyzed. Only semisolid samples were analyzed and show increasing values for hardness, adhesiveness, and compressibility. The loading of UA leads to an increase in values, corroborating the flow rheology measurements, since the hardness and compressibility were due to the attendant increased viscosities of these formulations [81,82]. Adhesiveness is more of a surface characteristic and depends on the combined effect of adhesive and cohesive forces [83]. It is also related to bioadhesive or mucoadhesive properties [84,85]. To achieve adequate efficacy and user acceptance of topical products, spreadability is an important property, related primarily to hardness and compressibility [86,87].

Bioadhesion can be defined as the state in which two materials, at least one of which is biological in nature, are maintained together for a prolonged time period by means of interfacial forces [88]. Prolonged efficacy via dermal administration of creams, solutions, and lotions is unexpected, since such preparations can be easily removed from the skin by moisture, temperature, and physical movements [89]. Therefore, the design of novel dosage formulations with bioadhesive properties is desirable for establishing long residence times at the application site and reducing product administration frequency [90]. Table 1 shows force of detachment values and establishes that liquid crystalline systems have more ability to adhere to the skin’s surface. This phenomenon can be explained by the viscosity of lyotropic liquid crystals and has already been reported by other authors for cubic and hexagonal mesophases [91,92].

UA has been studied for its anti-inflammatory activities [93] and has been shown to suppress the activation of nuclear factor kappa B (NF-κB), nuclear factor of activated T-cells (NF-AT), and activator protein 1 (AP-1) in lymphocytes of mice [94]. Treatment with ursolic acid leads to reduction in the pro-inflammatory cytokines interleukin 6 (IL-6) and interferon gamma (IFN-γ) [90], and in vivo anti-nociceptive activity has been demonstrated [95,96,97].

Figure 4 shows the weight of ear samples after treatment and the potential of topical ursolic acid administration. Formulations compared to dexamethasone showed no significant difference in the mean weight, with the exception of AAU. Free UA, vehicle (A), and untreated groups were significantly changed compared to the dexamethasone (*p* < 0.05) and naïve (*p* < 0.001) groups. The index of edema was measured as close to 100% for untreated, vehicle (A), and free UA, and these values show significant differences compared to dexamethasone. For the formulations, all demonstrated differences compared to the untreated group, but when compared to dexamethasone only BAU and AAU showed differences. The treatment showed an inhibition of ~60%, 50%, 40%, 35%, and 25% to AAU, BAU, CAU, DAU, and dexamethasone as the positive control, respectively.

Our previous study showed that curcumin loaded into lamellar and hexagonal mesophases exhibited an ability to reduce paw edema due to increased contact time with the skin surface, aiding the absorption of curcumin [46]. Another study examined the incorporation of *trans*-resveratrol into the lamellar/hexagonal mesophase; the maximal inhibition of inflammation was found to be between 63.4–27.4%, and these formulations improved bioadhesive properties [47]. Lamellar structures have the ability to deliver *trans*-resveratrol and reduce symptoms of UVB-irradiation-induced skin damage by inhibiting edema, neutrophil recruitment, lipid hydroperoxidation, and oxidative stress [48]. Cubic and hexagonal mesophases were also investigated in an aerosol-induced rat paw edema model, and the results showed reduced inflammation with celecoxib [98]. These data support the use of colloidal systems for delivery of drugs to ameliorate skin inflammation.

## 4. Conclusions

Colloidal systems containing OA, PPG-5-CETETH-20, and purified water were prepared for the incorporation of UA, a hydrophobic drug. These systems have the potential to meet some of the challenges involved in hydrophobic drug nanoformulations for topical cutaneous application. The rheological properties of microemulsions and liquid crystals showed pseudoplastic behavior and thixotropic properties desirable for effective topical use. In addition, oscillatory studies were performed and indicated that the microstructure self-assembled. TPA showed a viscosity dependence, and these systems showed bioadhesive ability, allowing for prolonged exposure upon topical application to skin and offering long-lasting activity. The pharmacological study demonstrated up to 65% inhibition of inflammation on mouse ears, with improved inhibition for those formulations that presented liquid crystal mesophases, suggesting these colloidal formulations hold great potential in the delivery of UA to the skin.

## Figures and Tables

**Figure 1 pharmaceutics-15-00366-f001:**
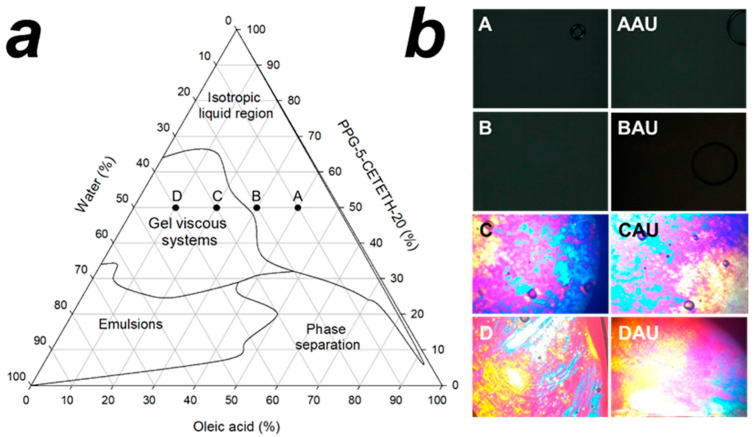
Phase ternary diagram of PPG-5-CETETH-20, oleic acid, and water mixture and their macroscopic visual characterization regions: inner symbols denoted the chosen formulations (**a**); polarized light microscopy obtained at 40× of magnification (**b**). A = formulation with OA%/S%/W% ratio of 40/50/10; B = formulation with OA%/S%/W% ratio of 30/50/20; C = formulation with OA%/S%/W% ratio of 20/50/30; D = formulation with OA%/S%/W% ratio of 10/50/40; AAU = formulation A loaded with AU; BAU = formulation B loaded with AU; CAU = formulation C loaded with AU; DAU = formulation Dloaded with AU.

**Figure 2 pharmaceutics-15-00366-f002:**
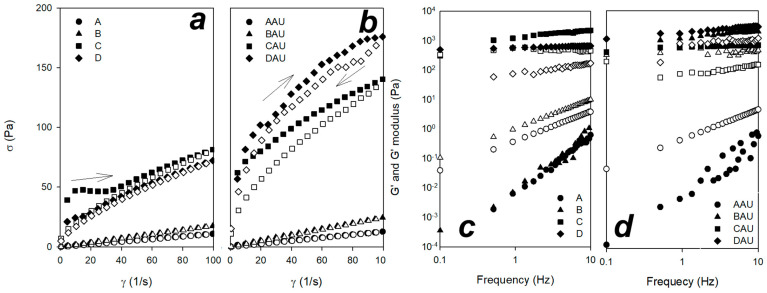
Rheograms obtained from unloaded and loaded formulations ((**a**,**b**), respectively). Ascending curves are denoted by close symbols and descending curves are denoted by open symbols; oscillatory sweep obtained from unloaded and loaded formulations ((**c**,**d**), respectively). G′ and G″ moduli are shown as closed and open symbols, respectively. A = formulation with OA%/S%/W% ratio of 40/50/10; B = formulation with OA%/S%/W% ratio of 30/50/20; C = formulation with OA%/S%/W% ratio of 20/50/30; D = formulation with OA%/S%/W% ratio of 10/50/40; AAU = formulation A loaded with AU; BAU = formulation B loaded with AU; CAU = formulation C loaded with AU; DAU = formulation Dloaded with AU.

**Figure 3 pharmaceutics-15-00366-f003:**
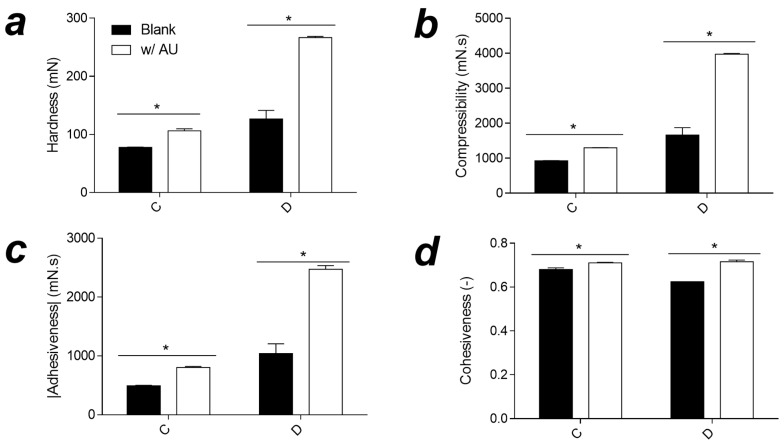
Hardness (**a**), adhesiveness (**b**), compressibility (**c**), and cohesiveness (**d**) obtained from C and D loaded and unloaded with ursolic acid samples. * *p* < 0.05 between same group. A = formulation with OA%/S%/W% ratio of 40/50/10; B = formulation with OA%/S%/W% ratio of 30/50/20; C = formulation with OA%/S%/W% ratio of 20/50/30; D = formulation with OA%/S%/W% ratio of 10/50/40.

**Figure 4 pharmaceutics-15-00366-f004:**
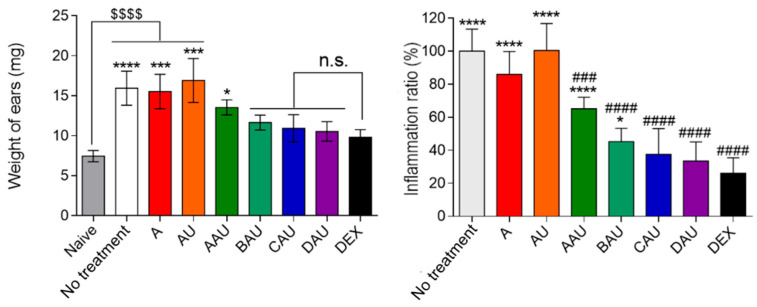
Anti-inflammatory activity of the colloidal systems with ursolic acid. The vehicle is the A formulation, and the control is an ointment containing dexamethasone 0.5% (*w*/*w*). Left graph shows weight of ears and right graph shows the inflammatory index (%) obtained. The data represent the mean ± S.D. of 5 mice. The statistical significances of ear thickness and inflammatory index were analyzed using variance analysis following Tukey’s multiple comparison test; * *p* < 0.05, *** *p* < 0.001, **** *p* ≤ 0.0001, and n.s. *p* > 0.05 compared to dexamethasone group; $$$$ is *p* ≤ 0.0001 compared to naïve group; ### *p* < 0.001 and #### *p* ≤ 0.0001 compared to no treatment group. A = formulation with OA%/S%/W% ratio of 40/50/10; B = formulation with OA%/S%/W% ratio of 30/50/20; C = formulation with OA%/S%/W% ratio of 20/50/30; D = formulation with OA%/S%/W% ratio of 10/50/40; AAU = formulation A loaded with AU; BAU = formulation B loaded with AU; CAU = formulation C loaded with AU; DAU = formulation D loaded with AU; DEX = dexamethasone.

**Table 1 pharmaceutics-15-00366-t001:** Values of constants obtained by Herschel–Bulkley model and Power Law model from rheology assays and bioadhesive measurement. AAU = A with A; BAU = B with A; CAU = C with A; DAU = D with A.

Sample	Herschel–Bulkley Model				Power Law Model			Bioadhesive Measurement
	*σ_o_* (Pa)	k (Pa.s^n^)	n	R^2^ Adjusted	S	n	R^2^ Adjusted	Force of Detachment (mN)
A	-	0.1469	0.9360	0.9999	0.0075	1.8735	0.9621	3.0 ± 1.0
AAU	1.0245	0.0890	1.0628	0.9991	0.0023	2.4691	0.7578	1.0 ± 1.5
B	-	0.2219	0.9501	1.0000	0.0010	3.0376	0.8452	4.0 ± 4.5
BAU	-	0.3777	0.9077	1.0000	1182.0945	0.2531	0.9348	3.0 ± 1.8
C	41.8661	0.0494	0.8606	0.9805	1239.1706	0.2676	0.9457	31.0 ± 6.4
CAU	52.0744	3.6331	0.6951	0.9991	581.3953	0.0808	0.8715	65.0 ± 18.5
D	15.3213	1.1716	0.8473	0.9971	575.5286	0.0625	0.9885	78.0 ± 7.7
DAU	6.5332	29.3480	0.3870	0.9928	1851.0950	0.2184	0.9704	33.0 ± 16.6

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
