# Peer review of "Design and Characterization of Lipid-Surfactant-Based Systems for Enhancing Topical Anti-Inflammatory Activity of Ursolic Acid"

_pharmaceutics, 2023, doi:10.3390/pharmaceutics15020366_

Round 1
Reviewer 1 Report
“Design and characterization of lipid-surfactant based-systems for enhancing topical anti-inflammatory activity of Ursolic acid” by Fonseca-Santos et al.
In this paper the authors have tested various colloidal systems for incorporation of the hydrophobic drug, ursolic acid. The main focus of this paper is the characterization of the liquid crystals and microemulsions. Pharmacological study of these colloidal systems has been limited. Results of one experiment has been presented and show some significant but not highly impressive inhibitory activity against inflammation. Overall, the paper has been well written and is an important contribution in the field. Here are some additional minor comments on the manuscript.
Line 26: Change “to delivery” to either “to deliver” or “for delivery of”
Line 31 – 33: “Ursolic acid…..rosemary” The language of this sentence matches exactly with that in Wikipedia. The authors should write it in their own language.
Line 63: “by mixing oil, OA,” This is misleading. Is OA the oil? If so, please change to “by mixing the oil, OA”. If there is a different oil besides OA, please specify that.
Line 70: “O%/S%/W%” The abbreviations O and S have not been mentioned before.
Line 160: “Groups 4 were” Do you mean, “Groups 4 – 7 were”?
Line 165: Change “ug/ml” to “mg/ml” (microgram/ml)
Line 175 “oedema” Use consistent spelling.
Lines 186-188: Please mention what AAU, BAU etc. represent. These abbreviations have been used throughout the manuscript but what they represent have not been mentioned.
Lines 218-219: The sentence is grammatically unclear.
Line 220: Change “due the” to “due to their”
Line 223: Change “was shown” to “as shown”
Line 223 and other places: “was” Not clear why all references to figures are in the past tense. It is true that experiments were done in the past but figures are there in the paper in the present.
Line 242: Change “values to close 1” to “values close to 1”
Line 247: Change “exbibit” to “exhibit”
Line 269: “oils an formulate” Not clear what this means. It appears ghat the sentence is missing a verb.
Line 274: “supramolecular self-assembled in formulations” Not clear what this means. Probably need to delete “in”
Line 275: “with increasing of polymer in solution” Not clear what this means. The authors probably mean “with increasing polymer concentration in solution”
Line 291: “to gives constants” Not clear what this means.
Line 352: Change “ghaph” to “graph”
Line 357, Fig 4: “No treated” in both graphs should read either “Not treated” or “No treatment”
Line 359: “These findings show that the liquid crystal was more effective for topical administration than microemulsions, possibly due to bioadhesive ability” True, that there is such a correlation, but one experiment cannot conclusively prove that.
Reviewer 2 Report
1. there are different ursolic acid based nano formulation reported in literature. author must update in this paper.
2. How the selection of oil and lipids have been done.
3. An 8-gram sample of each selected formulation. How much ursolic acid present.
4. is sink condition maintained for the study the release and permeation.
5. Statistical analysis must be added.
6. How the formulation applied to animals.
7. Experimental and introduction section needs extensive revision.
Round 2
Reviewer 2 Report
Accept